# Dynamic Sparse No Training ⊘:
# Training-Free Fine-tuning for Sparse LLMs

**Yuxin Zhang**[1,2†]    **Lirui Zhao**[1†]    **Mingbao Lin**[3]    **Yunyun Sun**[4]    **Yiwu Yao**[4]
**Xingjia Han**[4]    **Jared Tanner**[5]    **Shiwei Liu**[5,6,7]    **Rongrong Ji**[1,8‡*]

[1]Key Laboratory of Multimedia Trusted Perception and Efficient Computing,
  Ministry of Education of China, Xiamen University  [2] Pengcheng Lab  [3] Tencent Youtu Lab
[4]Huawei Technologies, [5]University of Oxford, [6]University of Texas at Austin
[7]Eindhoven University of Technology, [8]Institute of Artificial Intelligence, Xiamen University

## Abstract

The ever-increasing large language models (LLMs), though opening a potential path for the upcoming artificial general intelligence, sadly drops a daunting obstacle on the way towards their on-device deployment. As one of the most well-established pre-LLMs approaches in reducing model complexity, network pruning appears to lag behind in the era of LLMs, due mostly to its costly fine-tuning (or re-training) necessity under the massive volumes of model parameter and training data. To close this industry-academia gap, we introduce **Dynamic Sparse No Training (DS⊘T**[1]**)**, a **training-free** fine-tuning approach that slightly updates sparse LLMs without the expensive backpropagation and any weight updates. Inspired by the Dynamic Sparse Training, DS⊘T minimizes the reconstruction error between the dense and sparse LLMs, in the fashion of performing iterative weight pruning-and-growing on top of sparse LLMs. To accomplish this purpose, DS⊘T particularly takes into account the anticipated reduction in reconstruction error for pruning and growing, as well as the variance *w.r.t.* different input data for growing each weight. This practice can be executed efficiently in linear time since its obviates the need of backpropagation for fine-tuning LLMs. Extensive experiments on LLaMA-V1/V2, Vicuna, and OPT across various benchmarks demonstrate the effectiveness of DS⊘T in enhancing the performance of sparse LLMs, especially at high sparsity levels. For instance, DS⊘T is able to outperform the state-of-the-art Wanda by **26.79** perplexity at 70% sparsity with LLaMA-7B. Our paper offers fresh insights into how to fine-tune sparse LLMs in an efficient training-free manner and open new venues to scale the great potential of sparsity to LLMs. Codes are available at `https://github.com/zyxxmu/DSnoT`.

## 1 Introduction

Large language models (LLMs) (Zhang et al., 2022a; Touvron et al., 2023a; Brown et al., 2020) have recently emerged as the new favorite in various domains of natural language processing (NLP) (Wei et al., 2022b;a; Bubeck et al., 2023). Nevertheless, LLMs face a significant constraint: their extensive parameterization and computational demands present substantial challenges in terms of storage and deployment. For example, the GPT-175B model (Brown et al., 2020) eats up 320G of memory to load its parameters in FP16 precision, requiring at least five A100-80G GPUs for inference (Frantar & Alistarh, 2023). In response to this issue, there has been a surge of interest in compressing LLMs, as it holds the promise of LLMs while remarkably reducing memory usage and computational costs. To date, the majority of current effort for LLM compression falls into quantization (Yao et al., 2022; Lin et al., 2023; Frantar et al., 2022; Dettmers et al., 2023; 2022; Xiao et al., 2023; Shao et al., 2024; Ma et al., 2024), which compresses LLMs by diminishing the number of bits employed to represent weights or hidden states.

---

*†Equal contribution  ‡Corresponding author: rrji@xmu.edu.cn
[1]Pronounced "DS No T".

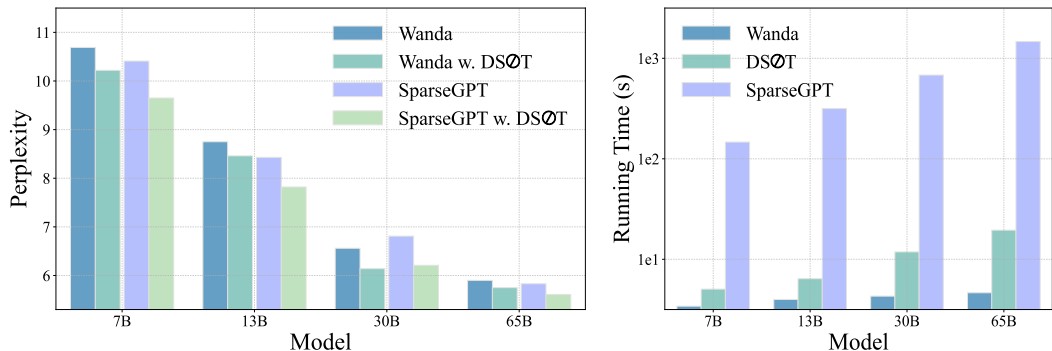

Figure 1: Perplexity on WikiText-2 (**left**) and running time (**right**) of different methods for pruning LLaMA-V1 model family at 60% sparsity rate. Without any training, DS⊘T consistently improves the performance of sparse LLMs, all within a linear time spectrum.

On the other hand, network pruning (LeCun et al., 1989; Han et al., 2015; Mocanu et al., 2018), a technique that removes superfluous weights to create a sparse and lightweight model, has received relatively little attention (Frantar & Alistarh, 2023; Sun et al., 2023). The plausible reason is that, network pruning usually appreciates at least one, usually many, iterations of fine-tuning or re-training to guarantee top performance (Frankle & Carbin, 2019; Yin et al., 2023). This fine-tuning step would cause a significant amount of compute and memory footprints due to the colossal model size and massive training data of modern LLMs, which even unnerves large corporations, let alone individual researchers.

Two previous arts have explored the possibility to scale pruning to billion-level LLMs without any fine-tuning. SparseGPT (Frantar & Alistarh, 2023) formulates LLM pruning as a layer-wise weight reconstruction problem, where the target falls into mitigating the output discrepancy, *w.r.t.*, reconstruction error, between dense and sparse LLMs. To solve the row-Hessian challenge, *i.e.*, the need for calculating the expensive inversion of a huge matrix for each row individually, SparseGPT iteratively applies OBS (Hassibi et al., 1993) to individually prune and updates weights in a column-wise manner, ultimately reaching the same optimal solution as applying the closed-form regression reconstruction. Wanda (Sun et al., 2023) proposes a new pruning metric that takes both weight magnitude and their corresponding input activations into consideration, performing on part with SparseGPT without the need for the expensive second-order information. The intuition behind Wanda lies in the existence of emergent outlier feature dimensions in large-scale LLMs which are significantly larger than typical features and meanwhile are essential for the optimal performance of LLMs (Dettmers et al., 2022). While these two approaches enable LLM pruning without performing fine-tuning, their performance is still far from satisfactory, *e.g.*, starting to lose performance at 20% sparsity with LLaMA-30B. *Therefore, it is imperative to enable fine-tuning for sparse LLMs to fully unlock the potential of sparsity to escalate the affordability of LLMs.*

In a parallel vein, Dynamic Sparse Training (DST), as outlined in previous research (Mocanu et al., 2018; Liu et al., 2019; Evci et al., 2020), has garnered considerable attention recently due to its significant saving potentials in the context of neural network training. Instead of training an entire network, DST selectively updates and maintains a subset of the network throughout the training process, while allowing the sparse network topology to dynamically evolve via a weight operation (Mocanu et al., 2018). Given its demonstrated efficacy in achieving efficient training, DST seems to be a promising candidate for efficient LLMs fine-tuning. However, it is essential to note that DST intrinsically requires the training of subnetworks via backpropagation, and the effectiveness of mask adaptation highly relies on a sufficient number of weight updates (Liu et al., 2021). Moreover, prior studies have indicated its failure when employed for fine-tuning small-scale BERT-level language models (Liu et al., 2023).

Fortunately, it is noteworthy that the pruning-and-growing step employed in DST solely stands as a training-free methodology, enabling sparse mask adaptation based on certain weight status, *e.g.,* magnitude (Mocanu et al., 2018). This offers an alternative perspective for addressing the aforementioned challenge: *While fine-tuning sparse LLMs through backpropagation can result in substantial computational overhead, we can explore the possibility of iteratively updating sparse mask in a training-free fashion as a viable alternative.* Based on this intuition, we introduce a **training-free**

fine-tuning approach – **Dynamic Sparse No Training (DS⊘T)**. This approach empowers the further refinement of sparse LLMs without any weight updates. To facilitate mask adaptation in favor of the sparse reconstruction problem, we propose new criteria for mask pruning and growing, by considering both the expectation and variance of the reconstruction error reduction when recovering a specific weight. It is worth emphasizing that the DS⊘T functions independently of the need for computationally intensive operations, such as gradient or Hessian matrices. Instead, it exclusively relies on a singular matrix multiplication operation to assess the reconstruction error.

We conduct comprehensive experiments to evaluate the effectiveness of DS⊘T with a variety of LLMs, including LLaMa-V1 (Touvron et al., 2023a) and LLaMa-V2 (Zhang et al., 2022a), Vicuna (Chiang et al., 2023), and OPT families (Zhang et al., 2022a), from 7 billion to 70 billion parameters. Our results demonstrate that DS⊘T consistently improves the performance of sparse LLMs by a good margin, especially at high sparsity levels > 50%. For instance, DS⊘T is able to improve the performance over Magnitude pruning, SparseGPT, and Wanda by 1.1e6, 4.31, and 1.87 perplexity with OPT-13B on WikiText-2 at 60% sparsity only using 7.3s on a single NVIDIA A100 GPU. Our work provides fresh insights in efficient sparse LLM fine-tune without weight updates and we hope to encourage more research in exploring benefits of sparsity in LLMs.

## 2 RELATED WORK

**Network Sparsification.** The process of eliminating redundant weights, known as network sparsification or network pruning, has served as a practical strategy to diminish the complexity of deep neural networks over the past decades (LeCun et al., 1989; Han et al., 2015). Despite the substantial body of literature, network pruning can be roughly classified based on the granularity of sparsity and the dependency of the pre-trained dense models. **I. Granularity of Sparsity:** The granularity of sparsity varies from coarse grains to fine grains. The coarse-grained granularity can be a group of weights (Gray et al., 2017; Ding et al., 2017), a complete neuron (Jiang et al., 2018); a filters/channels (Li et al., 2017), or an attention head (Voita et al., 2019), *etc*. On the other hand, fine-grained granularity eliminates the least important weights based on the selected criteria, regardless of where they are (Gale et al., 2019). The advantage of coarse-grained sparsity is its pronounced acceleration effect, which yet typically suffers from larger performance loss. Fine-grained sparsity enjoys performance superiority compared to other more structured forms of sparsity but receives limited support in common hardware. Nonetheless, recent advancements of dedicated fine-grained sparse patterns, such as *N:M* sparsity (Zhou et al., 2021; Zhang et al., 2022b), can be effectively accelerated. As such, this paper focuses on fine-grained network pruning. **II. Dependency of Pretrained Networks:** In parallel, sparsification techniques can be grouped into dense-to-sparse, and sparse-to-sparse methods based on the necessity of an over-parameterized dense network. The former entails embarking from a pre-trained dense model and discovering a sparse network (Han et al., 2015; Wen et al., 2016; Molchanov et al., 2017; Gale et al., 2019; Kurtic et al., 2022), usually followed by a retraining process to recover the optimal accuracy. On the other hand, sparse-to-sparse methods aim to train sparse neural networks from scratch, omitting any preliminary steps involving dense pre-training (Mocanu et al., 2018; Lee et al., 2019; Evci et al., 2020; Wang et al., 2020; Liu et al., 2021). Among them, **Dynamic Sparse Training (DST)** (Mocanu et al., 2018; Evci et al., 2020; Liu et al., 2021) stands out and receives upsurging interest due to its promise in saving both training and inference phases. In contrast to the conventional practices of pre-training followed by pruning, DST distinguishes itself by commencing with a randomly initialized sparse neural network. During a single training run, it dynamically adjusts the sparse network topology by such as pruning-and-growing, without the need for pre-training, while maintaining moderate training costs by, for example, keeping the similar sparsity ratios across all varying masks (Mostafa & Wang, 2019; Dettmers & Zettlemoyer, 2019; Yuan et al., 2021; Jayakumar et al., 2020).

While the crux of this paper focuses on the first category, *i.e.*, pruning a pre-trained LLM model, our proposed method is mainly inspired by the pruning-and-growing utilized in DST to iteratively refine the binary masks in a training-free manner, even though we do not conduct weight training as such. Another line of research, akin to our approach, demonstrates the existence of "**supermasks**" within randomly initialized network (Zhou et al., 2019; Ramanujan et al., 2020; Huang et al., 2022) or pre-trained networks (Mallya et al., 2018; Wortsman et al., 2020; Zhang et al., 2023), exhibiting the capacity to achieve commendable performance solely by seeking binary masks. However, it is imperative to note that these methods heavily rely on backpropagation, which is ill-suited for LLMs.

**Pruning of LLMs.** Compared to the well-established promise of pruning in pre-LLM small-scale models, the advancement of pruning in the context of LLMs appears to exhibit relatively modest progress. Firstly, traditional pruning generally requires at least one iteration of re-training to recover performance. Considering the substantial model size and massive datasets associated with LLMs, the prospect of conducting such resource-intensive re-training becomes a formidable challenge. To mitigate the above challenge, researchers have introduced pruning algorithms specifically devised for LLMs compression. Ma et al. (2023) explored structured sparse LLM by applying Taylor pruning (Molchanov et al., 2017) to remove entire weight rows, followed by the parameter efficient fine-tuning (PEFT) technique (Hu et al., 2021) fine-tuning. However, the fine-tuning phase still demands a considerable amount of data while the performance suffers a significant degradation, attributed primarily to the coarse-grained level of sparsity. Recent research endeavours have evolved towards the direction of unstructured pruning in one-shot without fine-tuning, demonstrating significant progresses. SparseGPT (Frantar & Alistarh, 2023) incorporates the Hessian inverse for pruning and subsequent residual weight updates, whereas Wanda (Sun et al., 2023) directly arrives at a sparse LLM model by a criterion depicted by the multiplication of the absolute values of weights and their activations with the aim to preserve outliers (Dettmers et al., 2022) emerged in LLMs. DS⊘T serves as an orthogonal perspective and can be organically integrated on top of them.

# 3 DYNAMIC SPARSE NO TRAINING – DS⊘T

**Preliminary.** LLM pruning entails the removal of a certain proportion of pre-trained weights to obtain a sparse LLM, with the objective of achieving minimal discrepancy between the output of the sparse and dense models (Hassibi et al., 1993). Solving this problem can be very arduous given the immense scale of LLMs. Therefore, it is more practical to formalize LLM pruning as a layer-wise reconstruction problem (Hubara et al., 2021; Frantar & Alistarh, 2023). Denote the weights of one dense LLM layer as $\mathbf{W} \in \mathbb{R}^{C_{\text{out}}, C_{\text{in}}}$, where $C_{\text{out}}$ and $C_{\text{in}}$ stand for the number of output and input channels respectively. Supposing we have $N$ calibration samples, the input activation can be represented as $\mathbf{A} \in \mathbb{R}^{C_{\text{in}}, N \times L}$ with $L$ be the sequence length. Pruning can be viewed as devising a binary mask $\mathbf{M} \in \{0,1\}^{C_{\text{out}}, C_{\text{in}}}$ to indicate whether weights are removed or not. Hence, the problem of LLM pruning given a specific pruning rate $p$ can be formalized as:

$$\min_{\mathbf{M},\mathbf{W}} \; ||\underbrace{\mathbf{W} * \mathbf{A} - (\mathbf{M} \odot \mathbf{W}) * \mathbf{A}}_{\Delta}||_2, \;\; s.t. \;\; 1 - \frac{||\mathbf{M}||_0}{C_{\text{out}} \cdot C_{\text{in}}} = p, \tag{1}$$

where $*$, $\odot$, $|| \cdot ||_2$ denote matrix multiplication, dot product operation, and $\ell_2$ norm, respectively. Note we refer $\Delta \in \mathbb{R}^{C_{out}, N \cdot L}$ as to the reconstruction error for ease of the following text.

**Dynamic Sparse No Training.** The problem defined in Eq. (1) can be addressed from two complementary perspectives. Firstly, it can be resolved through the initialization of sparse networks *i.e.*, devising criteria to prune weights that exhibit minimal impact on model output. For instance, SparseGPT (Frantar & Alistarh, 2023) employs second-order Hessian inverses, while Wanda (Sun et al., 2023) considers products of weight and activation norm as the guide for weight removal. Secondly, for the obtained sparse networks, the remaining weights can be naturally fine-tuned to further compensate for the reconstruction error (Han et al., 2015). Unfortunately, this requires substantial training resources, which is not practical given the large volumes of LLMs. Therefore, SparseGPT adjusts the remaining weights via an iterative OBS update (Hassibi & Stork, 1992), which as a consequence remarkably reduces the computing demands.

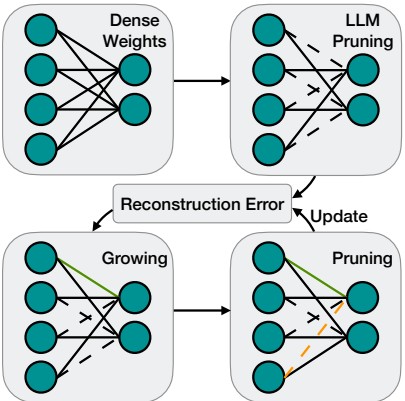

Figure 2: Framework of DS⊘T.

In this work, our focus is on the second part, *i.e.*, how to efficiently reduce the reconstruction error of a given pruned sparse network to its dense counterpart? Instead of fully fine-tuning (Han et al., 2015) or partially updating the pruned LLMs (Frantar & Alistarh, 2023) to recover performance, we introduce an ultra-efficient yet effective alternative to refine the sparse mask after pruning based on their

contribution to the reconstruction error. Our approach is inspired by the pruning-and-growing operation used in Dynamic Sparse Training (Mocanu et al., 2018; Evci et al., 2020). DST incorporates the processes of weight pruning and weight growing within the framework of sparse network training, contributing to the discovery of improved sparse topologies. Note that this pruning-and-growing operation solely serves as a training-free approach that is able to adapt sparse masks towards a desirable perspective, *e.g.,* loss minimization. Based on this insight, we propose **DS⊘T**, a training-free fine-tuning method for sparse LLMs that strips weights updating in DST and keeps the pruning-and-growing by converting the optimization objective to the reconstruction error of each weight row. We isolate pruning-and-growing from network training, and formulate it as an iterative approach to progressively optimize sparse masks towards the desirable ones achieving minimal reconstruction error represented by Eq. (1).

Specifically, DS⊘T starts with a sparse LLM which can be pruned by any existing criteria (Jaiswal et al., 2023; Sun et al., 2023; Frantar & Alistarh, 2023). Then, it performs iterative weight growing and pruning by looking at the reconstruction error as defined in Eq. (1), with especially-designed criteria to decrease the output discrepancy between sparse LLMs and their dense counterparts. The framework of DS⊘T is illustrated in Figure 2 and its main parts are detailedly described below.

**Growing Criterion.** As each output neuron is computed independently, we use one weight row $\mathbf{W}_r$ and the corresponding mask $\mathbf{M}_r$ for illustration. Given sparse weight row $\mathbf{M}_r \odot \mathbf{W}_r$, we attempt to revive pruned weight that leads to the most decrease on $\Delta_r$ across

---

**Algorithm 1:** Pseudocode of DS⊘T.

**Input:** A sparse layer with weight $\mathbf{W}\odot$, maximum cycle $T$, update threshold $\epsilon$.

**Workflow of DS⊘T:**

  Initialize reconstruction error $\Delta$ via Eq. (1)
  **for** $r = 1$ *to* $C_{out}$ **do**
    **for** $t = 1$ *to* $T$ **do**
      Obtain the growing index $i$ via Eq. (2).
      Obtain the pruning index $j$ via Eq. (3).
      $\mathbf{M}_{r,i} = 1$
      $\mathbf{M}_{r,j} = 0$
      Update reconstruction error $\Delta_r$ via Eq. (1).
      **if** $\Delta_r < \epsilon$ **then**
        **break**

  **return** Fine-tuned sparse weights $\mathbf{W} \odot \mathbf{M}$.

---

different input activations. Therefore, our growing criterion considers both the expectation and variance of the reconstruction error change when recovering a weight back. In particular, the index $i$ of the revived weights is derived as follows:

$$i = \begin{cases} \arg\max_{k} \neg\mathbf{M}_{r,k} \cdot \mathbf{W}_{r,k} \cdot \mathbb{E}[\mathbf{A}_r]/\mathrm{Var}(\mathbf{A}_r), & \text{if } \mathbb{E}[\Delta_r] > 0, \\ \arg\min_{k} \neg\mathbf{M}_{r,k} \cdot \mathbf{W}_{r,k} \cdot \mathbb{E}[\mathbf{A}_r]/\mathrm{Var}(\mathbf{A}_r), & \text{otherwise,} \end{cases} \tag{2}$$

where $\mathbb{E}(\cdot)$ and $\mathrm{Var}(\cdot)$ stand for the expectation and variance of given inputs across $N \times L$ different tokens. To explain, $\mathbb{E}[\mathbf{A}_r] \cdot \mathbf{W}_r$ represents the expected influence of weight growing on $\Delta_r$. Thus, based on the sign of the reconstruction error $\Delta_r$, we can determine which weight should be restored to approach the decrease of $\Delta_r$. Furthermore, we consider introducing the variance of the input activation to achieve a more robust revival. This is intuitive because if the influence of weight on $\Delta_r$ exhibits high variance across different inputs, restoring it may not result in stable error reduction.

**Pruning Criterion.** After choosing revived weights, we need to select another weight for pruning in order to maintain a fixed sparsity rate. However, the circumstances here are distinct: if we prune weights based on the impact of reconstruction error change as per Eq. (2), there is a risk of removing weights that significantly influence the output. This concern becomes especially critical when pruning LLMs due to the presence of emergent large magnitude features within them (Dettmers et al., 2022; Wei et al., 2022a; Schaeffer et al., 2023). To alleviate this, we utilize a transformed version of the Wanda metric (Sun et al., 2023). In addition to its standard criterion for pruning weights, we mandate that the selected weights should also contribute positively towards the reduction of reconstruction error when being pruned. This helps in preserving critical weights from removal without compromising the stable decrease of reconstruction error during the training-free fine-tuning process. Therefore, the pruning index $j$ is obtained as follows:

$$j = \begin{cases} \arg\min_{k,\mathbf{M}_{r,k} \cdot \mathbf{W}_{r,k} \cdot \mathbb{E}[\mathbf{A}_r] < 0} \mathbf{M}_{r,k} \cdot |\mathbf{W}_{r,k}| \cdot \|\mathbf{A}_r\|_2, & \text{if } \mathbb{E}[\Delta_r] > 0, \\ \arg\min_{k,\mathbf{M}_{r,k} \cdot \mathbf{W}_{r,k} \cdot \mathbb{E}[\mathbf{A}_r] > 0} \mathbf{M}_{r,k} \cdot |\mathbf{W}_{r,k}| \cdot \|\mathbf{A}_r\|_2, & \text{otherwise.} \end{cases} \tag{3}$$

Table 1: WikiText-2 Perplexity comparison for pruning LLMs at 60% sparsity rate.

| | LLaMA-V1 | | | | LLaMA-V2 | | | Vicuna | OPT |
|---|---|---|---|---|---|---|---|---|---|
| Method | 7B | 13B | 30B | 65B | 7B | 13B | 70B | 13B | 13B |
| Dense | 5.68 | 5.09 | 4.10 | 3.56 | 5.47 | 4.88 | 3.32 | 5.94 | 10.12 |
| Magnitude | 5.6e2 | 2.3e2 | 15.97 | 8.18 | 6.9e3 | 10.11 | 13.35 | 14.39 | 1.1e6 |
| w. DS⊘T | **66.70** | **30.71** | **10.81** | **7.37** | **40.01** | **9.41** | **6.77** | **12.02** | **2.4e2** |
| SparseGPT | 10.41 | 8.43 | 6.81 | 5.83 | 10.14 | 7.88 | 5.10 | 10.02 | 21.23 |
| w. DS⊘T | **9.65** | **7.73** | **6.69** | **5.64** | **9.67** | **7.57** | **5.07** | **9.38** | **16.92** |
| Wanda | 10.69 | 8.75 | 6.56 | 5.90 | 10.79 | 8.40 | 5.25 | 9.54 | 15.88 |
| w. DS⊘T | **10.22** | **8.46** | **6.44** | **5.75** | **10.59** | **8.18** | **5.20** | **9.18** | **14.01** |

**Workflow.** Given the criteria depicted above, the workflow of DS⊘T is outlined in Algorithm 1. In particular, it iteratively performs weight growing and pruning with respect to Eq. (2) and Eq. (3), with the reconstruction error updated until it reaches a pre-defined threshold. Meanwhile, we set a maximum pruning-and-growing cycle $T$ to prevent certain rows from being unable to reach the settled threshold $\epsilon$.

**Remark.** It's noteworthy that Algorithm,1 outlines the processing of each row in a sequential manner, primarily for the sake of simplicity. However, it's imperative to acknowledge that each row can, in fact, undergo parallel processing by employing a binary indicator to assess whether a particular row has satisfied the termination condition. Furthermore, the DS⊘T process eliminates the necessity for resource-intensive procedures such as backpropagation or the computation of gradient and Hessian matrices. Instead, it relies solely on several matrix multiplications to calculate the reconstruction error, a task that can be executed efficiently on GPUs. Subsequently, during each iteration of the DS⊘T process, the only operation is to update the reconstruction error through straightforward addition and subtraction operations during the pruning-and-growing process. This approach effectively circumvents the introduction of additional algorithmic complexity. In summary, DS⊘T preserves the simplicity associated with pruning LLMs, akin to the approaches employed in Wanda and Magnitude pruning.

## 4 EXPERIMENTAL RESULTS

### 4.1 SETTINGS

**Implementation details.** The implementation details of our proposed DS⊘T are presented as follows, mostly conforming to the existing setups (Frantar & Alistarh, 2023; Sun et al., 2023). In context to pruning configuration, we adhere to SparseGPT (Frantar & Alistarh, 2023), where a uniform sparsity is imposed for all layers with the first embedding layer and the final classification head skipped. Meanwhile, the calibration data consists of 128 segments, each with 2048 tokens. These segments are randomly selected from the first shard of the C4 dataset (Raffel et al., 2020). For the hyper-parameter settings, we set the maximum cycle $T = 50$ and the update threshold $\epsilon = 0.1$ in all experiments. Given sparse LLMs, we apply DS⊘T to fine-tune each layer in a progressive manner. We implement DS⊘T in PyTorch (Paszke et al., 2019) and use the HuggingFace Transformers library (Wolf et al., 2019) for handling models and datasets. All pruning experiments are conducted on NVIDIA A100 GPUs with 80GB of memory.

**Baselines.** We principally work with the LLaMA-V1 (Touvron et al., 2023a), LLaMA-V2 (Touvron et al., 2023b), Vicuna (Chiang et al., 2023), and OPT families (Zhang et al., 2022a), from 7 billion to 70 billion parameters, which are among the most powerful and open-source Large Language Models (LLMs) in the field today. We run DS⊘T on sparse LLMs pruned by various methods including (1) Magnitude-based pruning (Han et al., 2015) that discards weights based on their magnitudes. (2) SparseGPT (Frantar & Alistarh, 2023) that utilizes second-order Hessian inverses to ascertain unimportant weights. (3) Wanda (Sun et al., 2023) that removes weights with the smallest magnitudes multiplied by the corresponding input activation norms.

**Evaluation.** In accordance with prior studies (Frantar et al., 2022; Dettmers et al., 2023; Yao et al., 2022; Frantar & Alistarh, 2023), we assess the performance of pruned models by calcu-

Table 2: WikiText-2 perplexity performance of DS⊘T for fine-tuning sparse LLaMA-V1-7B/65B pruned by the Wanda metric at varying sparsity rates.

| | LLaMA-V1-7B | | | | | LLaMA-V1-65B | | | | |
|---|---|---|---|---|---|---|---|---|---|---|
| Sparsity | 50% | 60% | 70% | 80% | 90% | 50% | 60% | 70% | 80% | 90% |
| Wanda | 7.26 | 10.69 | 88.84 | 4.80e3 | 6.41e5 | 4.57 | 5.90 | 15.24 | 2.06e3 | 3.21e4 |
| w. DS⊘T | **7.12** | **10.22** | **62.05** | **4.12e3** | **8.43e4** | **4.54** | **5.75** | **12.93** | **1.82e3** | **2.09e4** |

lating the perplexity of language generation experiments on separate validation sets derived from WikiText2 (Merity et al., 2016). While perplexity has served as a stable and robust indicator of the generative performance of models (Dettmers & Zettlemoyer, 2023), we also examined the zero-shot capabilities of pruned models. In detail, we report the accuracy in six zero-shot tasks including PIQA (Bisk et al., 2020), StoryCloze (Mostafazadeh et al., 2017), ARC Easy and Challenge (Clark et al., 2018), HellaSwag (Zellers et al., 2019) and OpenBookQA (Mihaylov et al., 2018). We implement the lm-eval-harness (Gao et al., 2021) for the execution of all zero-shot tasks, with the report including both the accuracy results on each benchmark and overall average accuracy.

## 4.2 LANGUAGE MODELING

**Quantitative results.** The results for fine-tuning sparse LLM models at a uniform sparsity rate of 60% are presented in Table 1. Irrespective of the datasets used for evaluation, DS⊘T consistently delivers performance improvement for sparse LLMs with their original sizes varying from 7B to 70B. For instance, when pruning LLaMA-V1 with 7B parameters, DS⊘T is able to enhance the performance of Magnitude (Jaiswal et al., 2023), SparseGPT (Frantar & Alistarh, 2023), and Wanda (Sun et al., 2023) by 4.94e2, 0.76, and 0.47 perplexity on the Wikitext-2 validation sets, respectively. It is worth noting that, without any weight updating, DS⊘T consistently demonstrates better performance than SparseGPT, which requires expensive second-order Hessian inverses to update the sparse model. For larger models, the efficacy of DS⊘T is still hold with performance gain from 13.35 to 6.77 perplexity when fine-tuning sparse LLaMA-V2-70B obtained by magnitude pruning (Han et al., 2015). These findings suggest DS⊘T's versatility, being adaptable to boost the performance of sparse LLMs with different parameter budgets.

**Varying Sparsity Rates.** We further investigate the efficacy of DS⊘T when fine-tuning sparse LLMs with varying pruning rates. Table 2 shows that DS⊘T offers effective performance enhancement across various pruning methods at different sparsity levels. Particularly, this improvement becomes increasingly evident as the sparsity level grows.

Table 3: Time overhead (in seconds) for pruning LLaMA-V1 model family.

| Method | 7B | 13B | 30B | 65B |
|---|---|---|---|---|
| SparseGPT | 209 | 337 | 721 | 1285 |
| Wanda | 0.3 | 0.5 | 1.1 | 1.9 |
| Wanda+DS⊘T | 4.3 | 7.4 | 15.7 | 23.7 |

Table 4: Comparion with LoRA fine-tuning using 50% sparse LLaMA-7B.

| Method | Time Cost | Perplexity |
|---|---|---|
| Wanda+LoRA | 4h | 6.87 |
| Wanda+DS⊘T | 4.3s | 7.12 |

Table 5: Wikitext-2 perplexity comparison for pruning LLaMA-V1 model family with N:M pattern.

| Method | Sparsity | 7B | 13B | 30B | 65B |
|---|---|---|---|---|---|
| Dense | - | 5.68 | 5.09 | 4.10 | 3.56 |
| SparseGPT | 4:8 | 8.61 | 7.40 | 6.17 | 5.38 |
| w. DS⊘T | 4:8 | **8.32** | **7.05** | **6.10** | **5.12** |
| Wanda | 4:8 | 8.57 | 7.40 | 5.97 | 5.30 |
| w. DS⊘T | 4:8 | **8.45** | **7.25** | **5.91** | **5.26** |
| SparseGPT | 2:4 | 11.00 | 9.11 | 7.16 | 6.28 |
| w. DS⊘T | 2:4 | **10.03** | **8.36** | **6.82** | **5.80** |
| Wanda | 2:4 | 11.53 | 9.58 | 6.90 | 6.25 |
| w. DS⊘T | 2:4 | **10.89** | **9.05** | **6.76** | **6.14** |

**Computing efficiency.** We further demonstrate the efficiency of DS⊘T. Following Wanda, we only report the total pruning time and exclude the forward pass process shared by all methods. Table 3 compares the quantitative wall-clock overhead evaluated on NVIDIA A100 GPUs. It is indeed encouraging to observe that, as a fine-tuning approach, DS⊘T maintains a comparable computing time to Wanda, while demonstrating significantly higher efficiency compared to SparseGPT.

**Comparison with LoRA Fine-tuning.** To further demonstrate the ultra efficiency of DS⊘T in terms of fine-tuning, we also compare DS⊘T with parameter efficient fine-tuning (PEFT) method

Table 6: Zero-shot Accuracy comparison for pruning LLaMA-V1 model family at 60% sparsity rate.

| Params | Method | PIQA | HellaSwag | StoryCloze | ARC-e | ARC-c | OBQA | Mean |
|---|---|---|---|---|---|---|---|---|
| | Dense | 78.7 | 56.9 | 76.8 | 75.3 | 41.8 | 34.0 | 60.6 |
| 7B | SparseGPT | 73.1 | 44.8 | 71.5 | 62.6 | 30.2 | 24.4 | 51.1 |
| | **w. DS⊘T** | **73.7** | **47.2** | **72.3** | **62.8** | **30.9** | **29.4** | **52.7** |
| | Wanda | 73.0 | 43.6 | 69.7 | 62.8 | 30.3 | 25.0 | 50.7 |
| | **w. DS⊘T** | **73.2** | **43.7** | **70.0** | **63.6** | **30.8** | **25.8** | **51.2** |
| | Dense | 79.1 | 59.9 | 78.4 | 77.4 | 46.5 | 33.2 | 62.4 |
| 13B | SparseGPT | 75.6 | 49.0 | 74.8 | 68.4 | 36.2 | 27.6 | 55.2 |
| | **w. DS⊘T** | **75.8** | **51.5** | **75.8** | **69.8** | **36.3** | **28.8** | **56.3** |
| | Wanda | 74.9 | 48.9 | 74.5 | 68.9 | 34.9 | 27.6 | 54.9 |
| | **w. DS⊘T** | **75.0** | **49.1** | **75.1** | **69.2** | **35.4** | **28.0** | **55.3** |
| | Dense | 81.1 | 63.3 | 79.1 | 80.4 | 52.9 | 36.0 | 65.4 |
| 30B | SparseGPT | 76.8 | 55.0 | 78.4 | 74.7 | 43.3 | 32.2 | 60.1 |
| | **w. DS⊘T** | **77.3** | **58.0** | **78.8** | **74.8** | **45.6** | **32.8** | **61.2** |
| | Wanda | 77.7 | 56.7 | 79.1 | 76.2 | 46.5 | 31.6 | 61.3 |
| | **w. DS⊘T** | **78.1** | 56.7 | **79.7** | **76.8** | **46.6** | **32.6** | **61.7** |
| | Dense | 81.2 | 64.6 | 80.2 | 81.3 | 52.9 | 38.2 | 66.4 |
| 65B | SparseGPT | 79.6 | 58.3 | 80.5 | 77.4 | 46.6 | 33.4 | 62.6 |
| | **w. DS⊘T** | **79.9** | **59.8** | 80.4 | **78.1** | **46.9** | **34.6** | **63.3** |
| | Wanda | 79.9 | 58.9 | **80.6** | 78.2 | 47.1 | 34.8 | 63.3 |
| | **w. DS⊘T** | **80.9** | **59.6** | 80.2 | 78.2 | **47.7** | **36.0** | **63.7** |

LoRA (Hu et al., 2021). Table 4 presents a comparison of the time and performance of both methods in fine-tuning sparse LLaMA-7B. LoRA leverages the complete C4 dataset for a 5-hour fine-tuning and achieved a perplexity of 6.84. In stark contrast, DS⊘T only requires a brief duration of 4.3s and 128 samples to deliver a comparable performance, 7.12 perplexity. Taking into consideration the additional parameter burden incorporated by LoRA, the efficiency and practicality of DS⊘T is hold.

**N:M Fine-grained Sparsity**. Compared with unstructured sparsity, N:M fine-grained sparsity offers more practical speedup on the NVIDIA Ampere sparse tensor core (Nvidia, 2020). Thus, we also evaluate the effectiveness of DS⊘T on N:M fine-grained sparsity. Given the unique pattern of N:M sparsity that stipulates N non-zero components within M consecutive weight block, our implementation of DS⊘T involves a restriction on the position of pruning-and-growing weights. In particular, we select the pruned weight within the same block as the revived weight, thus the N:M characteristic is still maintained after fine-tuning. Table 5 lists the results for pruning LLaMA-V1 model family at 2:4 and 4:8 sparse patterns. Interestingly, even with the aforementioned extra restriction, DS⊘T can achieve more significant performance improvement compared to previous methods. For instance, when pruning LLaMA-V1 with 7B parameters, DS⊘T archives a perplexity of 10.89, enhancing Wanda (11.53) by a noticeable 0.64 ppl. Similar findings can be concluded when it comes to other models and sparse patterns. These results highlight the effectiveness of DS⊘T in boosting the performance of sparse LLMs, even with more complex sparsity constraints.

## 4.3 ZERO-SHOT TASKS

Following (Frantar & Alistarh, 2023; Sun et al., 2023), we also provided the accuracy performance of the LLaMA-V1 model family pruned at 50% sparsity rate on seven downstream zero-shot tasks. Averaging the accuracy over all tasks suggests DS⊘T's efficacy for enhancing sparse LLMs of any size. Particularly, DS⊘T improves the average accuracy of SparseGPT by 1.6% when pruning LLaMA-V1-7B (52.7% for DS⊘T and 51.1% for SparseGPT). For task-wise performance, DS⊘T is beneficial on all tasks, while there is not a fixed superiority for fine-tuning models obtained by different pruning methods. This phenomenon may evidence the reported relatively noisy evaluation results from these zero-shot experiments (Dettmers et al., 2022). However, the advantages of consistent performance improvement and efficiency of DS⊘T for zero-shot tasks are obvious.

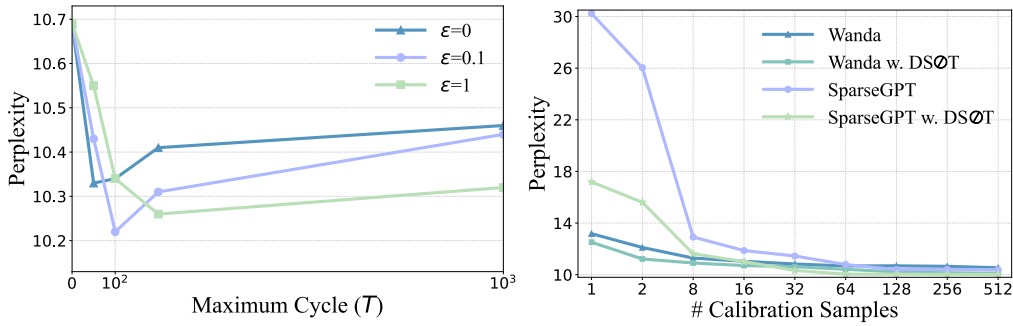

Figure 3: (**left**) Effect of the update schedule $(T, \epsilon)$ and (**right**) number of calibration sequences.

## 4.4 PERFORMANCE ANALYSIS

Next, we investigate the influence of the components within DS∅T, unfolds as its update schedule, pruning-and-growing criteria, and robustness to calibration samples. All experimental setups are based on the LLaMA-7B model pruned by the Wanda metric (Sun et al., 2023) with 60% sparsity.

**Update schedule.** In Figure 3 (left), we examine the performance of DS∅T under different hyper-parameter setting for the update schedule, including the maximum cycle $C$ and stop threshold $\epsilon$. The best performance is obtained with 50 cycles and 0.1 updating threshold. To analyze, smaller $C$ and larger $\epsilon$ both lead to an insufficient procedure for the decrease in reconstruction error. In contrast, running DS∅T without termination conditions also resulted in poor performance, most likely due to over-fitting of calibration data.

**Robustness to calibration samples.** In Figure 3 (right), we show the performance of pruning methods with varying numbers of sampled sequences for calibration. As can be observed, SparseGPT suffers serious performance degradation when calibration samples are limited, mostly due to the difficulty in estimating Hessian inverses in such cases. Fortunately, DS∅T consistently the performance of SparseGPT, even if only very few samples are given. These results further highlight the robustness of DS∅T for mitigating the reconstruction error.

**Pruning-and-growing criteria.** We further investigate the influence on criteria for prune and grow in Table 7. Note that when we transfer Eq. (2) to the prune criteria, the election of extreme values is also correspondingly reversed. As for the prune criterion, it can be seen that pruning weights that could bring the most reduction in reconstruction error actually led to a significant perfor-

Table 7: Effect of the pruning and growing criteria.

| Pruning \ Growing | $\lvert\mathbf{W}_{r,k}\rvert \cdot \lvert\lvert\mathbf{A}_r\rvert\rvert_2$ | Eq. (3) | Eq. (2) |
|---|---|---|---|
| $\lvert\mathbf{W}_{r,k}\rvert \cdot \lvert\lvert\mathbf{A}_r\rvert\rvert_2$ | 10.72 | 10.49 | 10.27 |
| Eq. (2) | 11.24 | 10.61 | 10.84 |
| Eq. (3) | 10.52 | 10.37 | **10.22** |

mance decrease. This indicates that while pursuing the reduction of reconstruction error, it is also essential to keep weights that exhibit an extremely large influence on the output, *e.g.*, weights within outlier channel. On the other hand, our proposed criteria based on the expectation and variance of the reconstruction error reduction achieved the best results among all growing criteria.

## 5 CONCLUSION

In this work, we introduce DS∅T, a training-free fine-tuning approach that enhances the performance of sparse LLMs without the expensive backpropagation or any weight updates. Taking inspiration from the success of sparse training in the pre-LLM pruning age, DS∅T adapts iterative weights growing and pruning in a sparse LLM, with a transferred target for minimizing the reconstruction error between dense and sparse LLMs outputs. To furnish guidance in the selection of weights to be pruned and grown, we introduce novel criteria that take into account the expectation and variance of the reconstruction error reduction by growing each weight concerning different inputs. Extensive experiments on pruning representative LLMs across various language benchmarks demonstrate the efficiency and effectiveness of DS∅T in boosting the performance of sparse LLMs.

ACKNOWLEDGEMENT

This work was supported by National Science and Technology Major Project (No. 2022ZD0118202), the National Science Fund for Distinguished Young Scholars (No.62025603), the National Natural Science Foundation of China (No. U21B2037, No. U22B2051, No. 62176222, No. 62176223, No. 62176226, No. 62072386, No. 62072387, No. 62072389, No. 62002305 and No. 62272401), and the Natural Science Foundation of Fujian Province of China (No.2021J01002, No.2022J06001).

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

# A    APPENDIX

## A.1    COMPLEMENTARY EXPERIMENTAL RESULTS

In this section, we supplement the main paper with more experimental outcomes, including a wider spectrum of results at varying sparsity rates, robustness analysis under random seeds, and quantitative comparison with varying numbers of calibration sequences.

**Varying Sparsity Rates.** This part delivers extended results of DS⊘T when fine-tuning sparse LLMs at alternating sparsity rates as a supplement to Section 4. The performance of various LLMs with sparsity rates oscillating between 10% and 90%, are presented in Table 8. Beneficial enhancements are consistently observable at all examined sparsity levels when employing DS⊘T, with the significance of improvements escalating concurrently with the increase in sparsity. It is noteworthy that the acceleration resultant from unstructured sparsity comes into play predominantly at high sparsity levels (exceeding 60%) Gale et al. (2020), thereby accentuating the indispensable efficacy of DS⊘T.

Table 8: WikiText-2 perplexity performance for fine-tuning LLMs at varying sparsity rates.

| Model | Method | 10% | 20% | 30% | 40% | 50% | 60% | 70% | 80% | 90% |
|---|---|---|---|---|---|---|---|---|---|---|
| LLaMA-V1-7B | Wanda | 5.70 | 5.82 | 6.00 | 6.39 | 7.26 | 10.69 | 88.84 | 4.80e3 | 6.41e5 |
| LLaMA-V1-7B | w. **DS⊘T** | **5.68** | **5.73** | **5.89** | **6.28** | **7.12** | **10.22** | **62.05** | **4.12e3** | **8.43e4** |
| LLaMA-V1-13B | Wanda | 5.10 | 5.13 | 5.25 | 5.51 | 6.15 | 8.75 | 55.89 | 3.66e3 | 1.54e6 |
| LLaMA-V1-13B | w. **DS⊘T** | **5.09** | **5.11** | **5.05** | **5.29** | **6.08** | **8.46** | **43.31** | **1.12e3** | **1.95e5** |
| LLaMA-V2-7B | Wanda | 5.49 | 5.59 | 5.74 | 6.06 | 6.92 | 10.79 | 75.01 | 2.36e3 | 7.87e3 |
| LLaMA-V2-7B | w. **DS⊘T** | **5.48** | **5.49** | **5.65** | **5.85** | **6.81** | **10.59** | **53.12** | **1.12e3** | **2.35e3** |
| LLaMA-V2-13B | Wanda | 4.91 | 4.99 | 5.13 | 5.37 | 7.88 | 8.30 | 46.05 | 1.06e3 | 1.22e5 |
| LLaMA-V2-13B | w. **DS⊘T** | **4.89** | **4.91** | **5.01** | **5.25** | **7.57** | **8.13** | **33.19** | **2.59e2** | **3.49e4** |
| OPT-13B | Wanda | 10.13 | 10.09 | 10.12 | 10.63 | 11.92 | 15.88 | 55.07 | 13722 | 7.61e5 |
| OPT-13B | w. **DS⊘T** | **10.12** | **10.08** | **10.11** | **10.41** | **11.28** | **14.01** | **45.10** | **8.43e3** | **2.33e5** |

**Varying Number of Sample Sequences.** Table 9 shows the quantitative results of different methods with varying numbers of calibrated sequences in complementary with Figure 3. Indeed, SparseGPT largely outperforms Wanda when the sample number starts to exceed 512. The performance gap gets larger with the length of 2048. It is worth mentioning that the efficacy of DS⊘T is indeed obvious when very limited numbers of calibration samples are given. Meanwhile, it is also encouraging to see that DS⊘T can consistently improve the performance of SparseGPT and Wanda even with 2048 calibrated sequences. This highlights the effectiveness of DS⊘T even when the pruning baseline is considerably strong, *i.e.*, SparseGPT with long input length.

Table 9: WikiText validation perplexity for different methods in pruning LLaMA-V1-7B at 50% sparsity with varying number of calibration sequences.

| Sample Length | 1 | 2 | 8 | 16 | 32 | 64 | 128 | 256 | 512 | 1024 | 2048 |
|---|---|---|---|---|---|---|---|---|---|---|---|
| Wanda | 13.18 | 12.11 | 11.29 | 11.04 | 10.83 | 10.68 | 10.69 | 10.65 | 10.54 | 10.68 | 10.77 |
| w. **DS⊘T** | **12.52** | **11.22** | **10.91** | **10.71** | **10.62** | **10.42** | **10.22** | **10.15** | **10.12** | **10.38** | **10.41** |
| SparseGPT | 30.23 | 26.04 | 12.92 | 11.87 | 11.45 | 10.79 | 10.40 | 10.41 | 10.39 | 9.93 | 9.99 |
| w. **DS⊘T** | **17.19** | **15.61** | **11.62** | **11.02** | **10.33** | **10.04** | **10.04** | **10.03** | **10.02** | **9.66** | **9.70** |

**Robustness Analysis.** We further perform a robustness analysis of DS⊘T. Given that the results in Table 1 is evaluated under a fixed calibration set, Table 10 show the results with different calibration sets under 5 random seeds. The variance across random seeds is very low, suggesting the stability of DS⊘T, corroborating its efficacy as a tool in fine-tuning sparse LLMs.

Table 10: WikiText validation perplexity for pruning LLaMA-V1 and LLaMA-V2 models at 60% sparsity. We report the mean and standard deviation under 5 random seeds.

| | LLaMA-V1 | | LLaMA-V2 | |
|---|---|---|---|---|
| Method | 7B | 13B | 7B | 13B |
| Dense | 5.68 (±0.00) | 5.09 (±0.00) | 5.47 (±0.00) | 4.88 (±0.00) |
| SparseGPT | 10.42(±0.04) | 8.43(±0.02) | 10.14 (±0.03) | 7.88(±0.01) |
| w. **DS⊘T** | **9.64(±0.03)** | **7.73(±0.02)** | **9.68(±0.03)** | **7.57(±0.01)** |
| Wanda | 10.69(±0.01) | 8.75(±0.01) | 10.79(±0.01) | 8.40(±0.01) |
| w. **DS⊘T**(±0.01) | **10.22(±0.01)** | **8.46(±0.01)** | **10.59(±0.01)** | **8.18(±0.01)** |

