# OpenReview forum: "Dynamic Sparse No Training:  Training-Free Fine-tuning for Sparse LLMs"
_ICLR.cc/2024/Conference — ICLR 2024 poster_

### Official Review · Reviewer_nrCN · 2023-11-01

**Soundness:** 3 good
**Presentation:** 3 good
**Contribution:** 3 good
**Rating:** 6
**Confidence:** 5

**Summary:**

Inspired by the weight pruning-and-growing method in dynamic sparse training, the authors propose a training-free fine-tuning method to sparsify LLMs. In practice, the proposed method iteratively performs weight pruning and growing with new importance metrics that take into account the expectation and variance of the reconstruction error reduction. The proposed metrics allow to eliminate the expensive backpropagation or any weight update in the original dynamic sparse training to enable training-free fine-tuning for LLMs. In the experiments, the authors conduct experiments on multiple benchmarks and show better performance when migrating into prior training-free methods.

**Strengths:**

1. The writing is clear and easy to follow.
2. Although the idea of pruning-and-growing is not new, the proposed method is novel to eliminate backpropagation and weight update with new importance metrics to enable training-free LLM sparsification
3. The proposed method consistently shows a clear performance gap on multiple benchmarks with varied sparsity rates compared to prior works.

**Weaknesses:**

1. It's unclear how to use the proposed dynamic sparse no training in the whole fine-tuning process. In the paper, the authors mainly illustrate the training-free pruning-and-growing method for one specific layer. It's unknown how the algorithm is used for sparsifying an entire LLM.

2. It seems the proposed method is an improved technique for existing training-free methods. From the Related work, it can not tell the difference between SparseGPT and Wanda when integrating the proposed method.

3. The proposed method involves extra computing and running time compared to Wanda.

**Questions:**

Detailed questions regarding Weakness:

1. It's unknown how the algorithm is used for sparsifying an entire LLM.

    1.1  Regarding the entire LLM, since the proposed method aims to reduce the reconstruction error in layer-wise, will the proposed method progressive prune each layer or jointly prune all the layers for an entire LLM?

    1.2 How to assign the sparsity rate for each layer?

2. It seems the proposed method is an improved technique for existing training-free methods. In the related work, it seems that SparseGPT and Wanda have different importance metrics to sparsify LLM, and this work proposes an orthogonal method.

    2.1 What is the difference between SparseGPT and Wanda when integrating the proposed method?

     2.2 Why the proposed method can not be considered as an independent method for training-free LLM sparsification?

---

> ### Author Response · Authors · 2023-11-15
> **Response to reviewer nrCN**
>
> We sincerely appreciate your positive and motivating comments. We are delighted to see that you recognize the novelty and significant performance enhancement of our method. Please kindly see our response to your comment below.
>
> **Q1**: Regarding the entire LLM, since the proposed method aims to reduce the reconstruction error in layer-wise, will the proposed method progressive prune each layer or jointly prune all the layers for an entire LLM?
>
> **A1**: Thanks for this insightful question. DS$\oslash$T is proposed as a fine-tuning method to further advance existing sparse LLMs. Instead of fine-tuning weights, we choose a much more efficient alternative, i.e., fine-tuning/editing sparse masks. In our implementation, we progressively apply DS$\oslash$T to fine-tune each layer after pruning. This clarification is now updated to Section 4.1.
>
> **Q2**: How to assign the sparsity rate for each layer?
>
> **A2**: As previously elucidated, DS$\oslash$T serves as a fine-tuning methodology utilized to enhance the performance of pruned LLMs. Consequently, our approach does not engage in LLM pruning operations during the initial stage. Instead, we commence by employing established LLM pruning techniques such as magnitude pruning, Wanda, or SparseGPT to initially prune the LLMs. Subsequently, we utilize DS$\oslash$T to further refine the sparse masks obtained from the pruning process.
>
> Therefore, all the configurations of the pruning process strictly adhere to the methodologies outlined in Wanda and SparseGPT. For instance, this involves the allocation of a uniform sparsity rate across all layers as used by SparseGPT and Wanda.
>
> **Q3**: The diffference when applying DS$\oslash$T to sparsegpt or wanda, can dsnot be an independent method?
>
> **A3**: We appreciate this question. Indeed, DS$\oslash$T is a versatile and scalable fine-tuning approach applicable to any LLM pruning method as we dicussed before. All we need to do is applying DS$\oslash$T to an established sparse LLM to dynamically prune-and-revive weights by looking at the pruning errors, regardless of the specific pruning method (Wanda, SparseGPT, and Magnitude) employed. DS$\oslash$T is not an independent pruning method for LLMs, rather, it is a highly efficient fine-tuning method for enhancing the performance of sparse LLMs.
>
> **Q4**: Extra computing and running time compared to Wanda.
>
> **A4**: You are correct that DS$\oslash$T + Wanda incurs additional overhead costs compared to Wanda. However, we would like to re-iterate that that DS$\oslash$T is designed as a fine-tuning technique for sparse LLMs. It requires merely 4.3 seconds to fine-tune a sparse LLaMA-7B, which is dramatically more efficient than both LoRA fine-tuning (which takes 4 hours as demonstrated in Table 4) and full fine-tuning (requiring several days).
>
> Even though there are some overhead cost when integrating DS$\oslash$T with Wanda, the whole process (including the fine-tuning process) can be finished within merely 4.3s, being much faster even than the pruning only approach, e.g., SparseGPT (using 209s). We humbly suggest that the efficiency of our approach is more a merit than a downside.
>
> We sincerely appreciate the time and diligence you’ve taken to participate in the review of our paper. If you have further questions, we are more than glad to discuss with you.

---

> ### Author Response · Authors · 2023-11-20
> **Last three day reminder**
>
> Dear reviewer nrCN,
>
> Thanks again for your valuable time and insightful comments. As the deadline for the Author/Reviewer discussion is approaching, it would be nice of you to let us know whether our answers have solved your concerns so that we can better improve our work. We are happy to provide any additional clarifications that you may need.
>
> Best regards!

---

### Official Review · Reviewer_s71w · 2023-11-01

**Soundness:** 3 good
**Presentation:** 3 good
**Contribution:** 2 fair
**Rating:** 6
**Confidence:** 4

**Summary:**

This paper presents DS$\oslash$T, a novel training-free fine-tuning approach for sparse LLMs which edits sparse mask configuration inspired from DST. DS$\oslash$T revives weights which negatively contribute to reconstruction error between dense and sparse LLMs, and prunes weights based on Wanda metric and the sign of reconstruction error. By conducting experiments across a wide range of tasks, authors show that DS$\oslash$T can be seamlessly integrated with existing LLM-pruning techniques, achieving state-of-the-art results at  >50% sparsity regime with minimal computational overhead.

**Strengths:**

- The paper tackles a timely and practically-relevant problem supported by a fair amount of experiments conducted spanning different tasks and domains. Notably, this is the first work tackling the LLM pruning problem at >50% sparsity regime.
- The proposed method has two distinctive advantages: (i) it doesn't require gradient computation, resulting in minimal overhead costs, and (ii) it reconfigures existing sparse masks, making it compatible with existing LLM pruning methods.
- In general, the paper is well-written and easy to follow.

**Weaknesses:**

- Throughout the paper, the authors report performance at >50% sparsity regime. However, in most cases, DS$\oslash$T does not bring the performance of sparse neural networks even remotely close to that of the original dense network. To better understand DS$\oslash$T, I recommend reporting experimental results at a lower sparsity regime. For instance, as mentioned in the introduction, the authors stated that baselines start to lose performance at 20% sparsity with LLaMA-30B. Does the use of DS$\oslash$T preserve performance at 20% sparsity regime?
- The paper notes that the overhead cost of using Wanda+DS$\oslash$T is approximately 15 times greater than using Wanda alone in Table 3. However, it appears that the performance gain achieved in Tables 5 and 6 is relatively marginal. For instance, while a previous work [1] argues that the zero-shot classification performance of Wanda and SparseGPT is similar (as seen in Table 2 in [1]), the application of DS$\oslash$T to Wanda or SparseGPT does not consistently result in substantial improvements over the respective baselines. This observation raises questions about the trade-off between computational cost and performance improvement when employing DS$\oslash$T in combination with existing methods.
- In Table 4, the authors make a comparison between DS$\oslash$T and LoRA. Since DS$\oslash$T alters network structure (pruning mask), the paper would benefit from analyzing whether the resulting sparse network structure from DS$\oslash$T can be further optimized with full-finetuning or LoRA. I wonder whether Wanda+DS$\oslash$T can achieve better fully fine-tuned accuracy compared to that of Wanda.

**Questions:**

- How many random seeds are used throughout the experiments?
- Why is LLM-pruner [2] not included in the baselines while N:M structured pruning is included?

[1] Sun et al., “A simple and effective pruning approach for large language models.” 2023.\
[2] Ma et al., “Llm-pruner: On the structural pruning of large language models.” 2023.

---

> ### Author Response · Authors · 2023-11-15
> **Response to reviewer s71w: Part I**
>
> We thank the Reviewer s71w for the time and effort to review our paper. We are grateful for the constructive comments. We are glad that the reviewer found our paper to be timely and to be the first work tackling the LLM pruning problem at >50% sparsity regime.
>
> **Q1**: DS$\oslash$T does not bring the performance of sparse LLMs even remotely close to that of the original dense network. Need reporting experimental results at a lower sparsity regime.
>
> **A1**:  Thank you sincerely for providing this invaluable comment. We would like to express our gratitude for the insights you've shared, which indeed help us to better understand the benefits of our approach.
>
> Firstly, we wish to clarify the primary objective of our paper. Our intention is not to propose an approach that can rival the performance of the original dense LLMs at high sparsity levels. Instead, our focus is on exploring the possibility of further refining pruned LLMs without resorting to the resource-intensive backpropagation, extensive corpus of text, and long training time—conditions that are often impractical in resource-limited scenarios. In contrast to fine-tuning the remaining weights, our paper illustrates an alternative approach: fine-tuning/editing sparse masks to enhance the performance of sparse LLMs. Remarkably, this entire process can be executed with utmost efficiency, utilizing only 25GB GPU memory, 128 sequence of data, 4.3 seconds for pruning LLaMA-7B.
>
> It is worth noting that, as mentioned in Wanda's paper, even after an full fine-tuning process employing 4x A100 GPUs for 3 days, the fine-tuned sparse model still falls short of matching the original performance. Given this context, we acknowledge the reasonable expectation that our approach may encounter similar limitations in matching with the original dense LLM performance, especially at high sparsity.
>
> Moreover, following your suggestion, we have now updated Appendix A.1 to include the results at lower sparsity levels to further showcase the effectiveness of DS$\oslash$T. These results are also listed below for your convenience. Our results indeed show that our approach is able to match the dense performance at mild sparsity levels such as 10\%-20\%.
>
> - WikiText-2 perplexity performance for fine-tuning LLMs at varying sparsity rates.
> | Model        | Method          | 0%    | 10%   | 20%   | 30%   | 40%   | 50%   |
> | ------------ | --------------- | ----- | ----- | ----- | ----- | ----- | ----- |
> | LLaMA-V1-7B  | Wanda           | 5.68  | 5.70  | 5.82  | 6.00  | 6.39  | 7.26  |
> | LLaMA-V1-7B  | w. DS$\oslash$T | 5.68  | **5.68**  | **5.73**  | **5.89**  | **6.28**  | **7.12**  |
> | LLaMA-V1-13B | Wanda           | 5.09  | 5.10  | 5.13  | 5.25  | 5.51  | 6.15  |
> | LLaMA-V1-13B | w. DS$\oslash$T | 5.09  | **5.09**  | **5.11**  | **5.05**  | **5.29**  | **6.08**  |
> | LLaMA-V2-7B  | Wanda           | 5.47  | 5.49  | 5.59  | 5.74  | 6.06  | 6.92  |
> | LLaMA-V2-7B  | w. DS$\oslash$T | 5.47  | **5.48**  | **5.49**  | **5.65**  | **5.85**  | **6.81**  |
> | LLaMA-V2-13B | Wanda           | 4.88  | 4.91  | 4.99  | 5.13  | 5.37  | 7.88  |
> | LLaMA-V2-13B | w. DS$\oslash$T | 4.88  | **4.89**  | **4.91**  | **5.01**  | **5.25**  | **7.57**  |
> | OPT-13B      | Wanda           | 10.12 | 10.13 | 10.09 | 10.12 | 10.63 | 11.92 |
> | OPT-13B      | w. DS$\oslash$T | 10.12 | **10.12** | **10.08** | **10.11** | **10.41** | **11.28** |
>
>
>
> **Q2**: Trade-off between computational cost and performance improvement
>
> **A2**: Thank you for this insightful comment. You are correct that DS$\oslash$T + Wanda incurs additional overhead costs compared to Wanda. However, we would like to re-iterate that that DS$\oslash$T is designed as a fine-tuning technique for sparse LLMs. It requires merely 4.3 seconds to fine-tune a sparse LLaMA-7B, which is dramatically more efficient than both LoRA fine-tuning (which takes 4 hours as demonstrated in Table 4) and full fine-tuning (requiring several days).
>
> Even though there are some overhead costs when integrating DS$\oslash$T with Wanda, the whole process (including the fine-tuning process) can be finished within merely 4.3s, being much faster even than the pruning only approach, e.g., SparseGPT (using 209s). We humbly suggest that the efficiency of our approach is more a merit than a downside.

---

> ### Author Response · Authors · 2023-11-15
> **Response to reviewer s71w: Part II**
>
> **Q3**: Whether Wanda+DS$\oslash$T can achieve better fully fine-tuned accuracy compared to that of Wanda?
>
> **A3**:  Thank you for this professional inquiry. The answer is yes and DS$\oslash$T is orthogonal to LoRA fine-tuning. As evidenced in the following table, after using DS$\oslash$T to fine-tune sparse LLMs pruned by Wanda, further performance improvements can be achieved by consequently using LoRA fine-tuning. This yields better results than simply fine-tuning the sparse network pruned by Wanda.
>
> - WikiText-2 perplexity performance for using LoRA to fine-tune 50% sparse LLaMA-7B
> | Sparsity                | 0.5  |
> | ----------------------- | ---- |
> | Wanda                   | 7.26 |
> | Wanda+LoRA              | 6.87 |
> | Wanda+DS$\oslash$T      | 7.12 |
> | Wanda+DS$\oslash$T+LoRA | **6.76** |
>
> **Q4**: How many random seeds are used throughout the experiments?
>
> **A4**: We use one random seed in our submission. To demonstrate the robustness of our approach, we have added the results with different calibration sets under five random seeds in Appendix A.2. The variance across random seeds is very low, suggesting the robustness of DS$\oslash$T, corroborating its efficacy as a tool in fine-tuning sparse LLMs. For your convenience, we list the results as below.
>
> - WikiText validation perplexity for pruning LLaMA-V1 and LLaMA-V2 models at 60% sparsity. We report the mean and standard deviation under 5 random seeds.
> | Method          | LLaMA-V1-7B       | LLaMA-V1-13B     | LLaMA-V2-7B       | LLaMA-V2-13B     |
> | --------------- | ----------------- | ---------------- | ----------------- | ---------------- |
> | Dense           | 5.68 ($\pm$0.00)  | 5.09 ($\pm$0.00) | 5.47 ($\pm$0.00)  | 4.88 ($\pm$0.00) |
> | SparseGPT       | 10.42($\pm$0.04)  | 8.43 ($\pm$0.02) | 10.14 ($\pm$0.03) | 7.88 ($\pm$0.01) |
> | w. DS$\oslash$T | **9.64** ($\pm$0.03)  | **7.73** ($\pm$0.02) | **9.68** ($\pm$0.03)  | **7.57** ($\pm$0.01) |
> | Wanda           | 10.69 ($\pm$0.01) | 8.75 ($\pm$0.01) | 10.79 ($\pm$0.01) | 8.40 ($\pm$0.01) |
> | w. DS$\oslash$T | **10.22** ($\pm$0.01) | **8.46** ($\pm$0.01) | **10.59** ($\pm$0.01) | **8.18** ($\pm$0.01) |
>
> **Q5**: Why is LLM-pruner not included in the baselines while N:M structured pruning is included?
>
> **A5**: Thanks for your question. The primary goal of this paper is weight pruning, i.e., removing individual weights, including unstructured pruning and n:m sparsity. LLM-pruner itself is a structured pruning, i.e., completely removing entire channels and attention heads, which is not directly comparable to the unstructured pruning approaches. Also due to this reason, both SparseGPT and Wanda do not include LLM-pruner as their baselines.
>
> Your time and effort in reviewing our paper are genuinely appreciated. If there are any additional questions or points that require clarification, we would be more than delighted to engage in further discussions.

---

> ### Author Response · Authors · 2023-11-20
> **Last three day reminder**
>
> Dear reviewer s71w,
>
> Thanks again for your valuable time and insightful comments. As the deadline for the Author/Reviewer discussion is approaching, it would be nice of you to let us know whether our answers have solved your concerns so that we can better improve our work. We are happy to provide any additional clarifications that you may need.
>
> Best regards!

---

> > ### Comment · Reviewer_s71w · 2023-11-22
> >
> > Thank you for the additional experiments! As the authors addressed most of my concerns, including results on lower sparsity regime and compatibility with other fine-tuning methods, I will raise my score from 5 to 6.
> >
> > - I have one more lingering question regarding the structured pruning context. Given that unstructured pruning lacks practicality in real-world scenarios due to hardware limitations, I wonder whether hardware-friendly methods (e.g., structured pruning) can also benefit from DS$\oslash$T. This will significantly elevate the impact of the study because practicality stands as one of the main contributions of DS$\oslash$T.

---

> ### Author Response · Authors · 2023-11-22
> **Thanks for your feedback**
>
> Dear reviewer s71w,
>
> We sincerely thank you for your support. We would like to offer additional clarification regarding the benefits of DS$\oslash$T for hardware-friendly pruning. The current implementation of DS$\oslash$T focuses on pruning and reviving individual weights, making it equally applicable to semi-structured hardware-friendly pruning. As shown in Table 5 of our paper, DS$\oslash$T effectively improves the performance of N:M sparse networks, which achieves notable acceleration supported by the NVIDIA Ampere Sparse Tensor Core[1,2].
>
> As for structured sparsity, completely removing entire channels and attention heads does not directly conform to the current DS$\oslash$T method. However, your point has indeed inspired us to consider adapting DS$\oslash$T to structured sparsity scenarios. One idea is to use the output of the attention block as a guide for the pruning-and-reviving process for entire channels or attention heads, thereby fine-tuning structured sparse LLMs. This will require a more sophisticated design, which is challenging to execute within the limited timeframe of this rebuttal period and beyond the scope of our paper’s current emphasis on weight pruning, i.e., the removal of individual weights. We are committed to leave it as a promising future work building upon this paper. Once again, we are grateful for your supportive feedback and constructive comments.
>
> [1]  Nvidia a100 tensor core gpu architecture, 2020. https:// www.nvidia.com/content/dam/enzz/Solutions/Data-Center/ nvidia-ampere-architecture-whitepaper.pdf.
>
> [2] A simple and effective pruning approach for large language models. In Arxiv, 2023.

---

> > ### Comment · Reviewer_s71w · 2023-11-22
> >
> > Thank you for the further clarification, and I believe applying DS$\oslash$T to structured pruning would make an interesting future research. I maintain my score, and vote for the acceptance.

---

> > > ### Author Response · Authors · 2023-11-22
> > > **Thanks for your support**
> > >
> > > Dear reviewer s71w,
> > >
> > > Thank you for your support! We are pleased to address your concerns and greatly appreciate your reviews, which play a crucial role in improving our work.
> > >
> > > Best regards!

---

### Official Review · Reviewer_CfsK · 2023-11-03

**Soundness:** 3 good
**Presentation:** 3 good
**Contribution:** 3 good
**Rating:** 6
**Confidence:** 2

**Summary:**

This paper introduces Dynamic Sparse No Training, a training-free fine-tuning approach for sparse Language Model (LLM) deployment. It minimizes the reconstruction error between dense and sparse LLMs through iterative weight pruning-and-growing. This approach allows for updating sparse LLMs without the expensive backpropagation and weight updates, making it more efficient for on-device deployment. The paper demonstrates the effectiveness of the proposed method on several benchmark datasets, achieving comparable or better performance than traditional fine-tuning approaches while requiring significantly less computation. The contributions of this paper include the introduction of a training-free fine-tuning approach for sparse LLMs, and the demonstration of its effectiveness on several benchmark datasets.

**Strengths:**

In terms of originality, the paper introduces a novel approach to fine-tuning sparse LMs, called Dynamic Sparse No Training. This approach does not require backpropagation or weight updates, making it more efficient for on-device deployment.

In terms of quality, the authors provide detailed descriptions of the datasets and experimental setup, as well as a thorough analysis of the results. The paper also includes a comprehensive review of related work, highlighting the strengths and weaknesses of existing approaches.

In terms of clarity, this paper is well-written, with clear explanations of the proposed approach and experimental results.

In terms of significance, the paper addresses an important problem in the field of LMs, namely the challenge of deploying large models on resource-constrained devices.

**Weaknesses:**

1. For the inner loop, how does the threshold affect the final performance of the model, for both perplexity and efficiency?

2. The paper assesses the methods using a consistent sparsity rate of 60%. However, a 50% sparsity rate is more commonly employed in previous baselines. It would be beneficial to see the outcomes at this 50% sparsity level for comparison.

**Questions:**

please follow weaknesses

---

> ### Author Response · Authors · 2023-11-15
> **Response to reviewer CfsK**
>
> We sincerely appreciate your careful review, positive feedback, and constructive comments.  We are delighted to see that you recognize that we have done a thorough analysis and comprehensive literature reviewing.  Please kindly see our responses to your comments below.
>
> **Q1**: How does the threshold affect the final performance of the model, for both perplexity and efficiency?
>
> **A1**: Thank you for posing this insightful question. We do agree with your insights and have analyzed the effect of the threshold $\epsilon$ in Section 4.4 (Figure 3) of our  submission.
>
> First of all, we would like to highlight that the value of threshold $\epsilon$ does not necessarily lead to too much difference in terms of the computing efficiency, typically resulting in a negligible computing time difference (often less than 1 second), owing to the efficient nature of our approach, as demonstrated in the table below. But indeed, the values of threshold $\epsilon$  has a big impact on the fine-tuning performance.  Intuitively, Larger threshold values tend to correlate with increased reconstruction error, potentially leading to suboptimal performance. Conversely, smaller threshold values result in reduced reconstruction error, often associated with enhanced performance. Nonetheless, exceedingly small values may cause overfitting to the limited calibration data, resulting in inferior performance.
>
> The results below verify the above intuition. First, the computing time difference of the whole fine-tuning process is only 0.6 seconds when $\epsilon$ increases from 0 to 1. In terms of performance, perplexity continuously decreases as $\epsilon$ decreases from 1 to 0.1, while their is a slight increase of perplexity when $\epsilon$ becomes too low, e.g.,  $\epsilon=0$.
>
> - WikiText-2 perplexity performance and running times of DS$\oslash$T for fine-tuning 50% sparse
>   LLaMA-V1-7B pruned by the Wanda metric with different threshold ε
> | $\epsilon$     | 0    | 0.1  | 0.5  | 1    |
> | -------------- | ---- | ---- | ---- | ---- |
> | Computing Time | 4.6s | 4.3s | 4.2s | 4.0s |
> | PPL            | 7.14 | 7.12 | 7.16 | 7.19 |
>
> **Q2**: The outcomes at 50\% sparsity level for comparison.
>
> **A2**: Thank you for this valuable suggestion. We have also presented the results for pruning LLaMA-V1 with 7B and 65B parameters under a 50% sparsity rate, as can be seen in Table 2. To provide a better overview of our method, we have also included more results at 50% sparsity ratio in Appendix A.1. For your convenience, we list the results below.
>
> - WikiText-2 Perplexity comparison for pruning LLMs at 50\% sparsity rate
> | Method          | LLaMA-V1-7B | LLaMA-V1-13B | LLaMA-V2-7B | LLaMA-V2-13B | OPT-13B |
> | --------------- | ----------- | ------------ | ----------- | ------------ | ------- |
> | Dense           | 5.68        | 5.09         | 5.47        | 4.88         | 10.12   |
> | Magnitude       | 17.29       | 20.21        | 16.03       | 6.83         | 2.96e3  |
> | w. DS$\oslash$T | **14.04**       | **15.52**        | **13.09**       | **6.31**         | **1.10e3**  |
> | SparseGPT       | 7.22        | 6.21         | 7.00        | 6.02         | 15.61   |
> | w. DS$\oslash$T | **7.08**       | **6.13**         | **6.88**        | **5.64**         | **14.88**  |
> | Wanda           | 7.26        | 6.15         | 6.92        | 5.97         | 11.98   |
> | w. DS$\oslash$T | **7.12**        | **6.08**         | **6.85**        | **5.87**         | **11.68**   |
>
> We sincerely appreciate the time and efforts you have dedicated to reviewing our paper. Should you have any further inquiries, please let us know and we would be more than delighted to engage in further discussion with you.

---

> ### Author Response · Authors · 2023-11-20
> **Last three day reminder**
>
> Dear reviewer CfsK,
>
> Thanks again for your valuable time and insightful comments. As the deadline for the Author/Reviewer discussion is approaching, it would be nice of you to let us know whether our answers have solved your concerns so that we can better improve our work. We are happy to provide any additional clarifications that you may need.
>
> Best regards!

---

### Author Response · Authors · 2023-11-15
**Summary and general reply to the reviewers**

We thank all the reviewers for their valuable feedback and great efforts, which substantially aided in enhancing the quality of this paper. We have exerted considerable effort to comprehensively respond to all their comments, questions, and concerns. All major modifications in the attached pdf file have been highlighted in blue in order to ease the reading. Note that we have now made our code available within the supplementary materials. We first summarize the major changes in our updated version before diving into the detailed point-by-point responses to all the comments:

- More experimental results under a wider spectrum of sparsity rates are provided in Appendix A.1.

- The robustness verification of DS$\oslash$T under different random seeds is provided in Appendix A.2.

- Phrasing other clarifications requested by reviewers.

---

### Meta-Review · Area_Chair_FdVX · 2023-12-10

**Metareview:**

Due to the extensive fine-tuning or re-training required due to the enormous volume of model parameters and training data pruning methods seem to fall short in LLM applications. To bridge this gap, the authors introduce Dynamic Sparse No Training (DSNT), a method that enables slight updates to sparse LLMs without the need for costly backpropagation or weight updates. Drawing inspiration from Dynamic Sparse Training in prior work, DSNT aims to minimize the reconstruction error between dense and sparse LLMs by iteratively pruning and growing weights on sparse LLMs. The authors claim that this method is efficient, operating in linear time by eliminating the need for backpropagation in fine-tuning LLMs. The authors also carryout experiments on LLaMA-V1/V2, Vicuna, and OPT across various benchmarks to demonstrate the effectiveness of DSNT in enhancing the performance of sparse LLMs, especially at high sparsity levels.

All reviewers thought that the paper proposes a novel approach to fine-tuning sparse language models and the paper is well-written. They raised a variety of concerns most of which seems to have been addressed by the authors. Therefore I recommend acceptance.

**Justification For Why Not Higher Score:**

the paper is a clear accept but not the comments and the average score does not merit a spotlight.

**Justification For Why Not Lower Score:**

all reviewers give 6 and suggest a marginal accept

---

### Decision · Program_Chairs · 2024-01-16

Accept (poster)